# Generalization Task for Developing Social Problem-Solving Skills among Young People with Autism Spectrum Disorder

**DOI:** 10.3390/children9020166

**Published:** 2022-01-28

**Authors:** Saray Bonete, Clara Molinero, Adrián Garrido-Zurita

**Affiliations:** 1Departament of Psychology, Universidad Francisco de Vitoria, 28223 Pozuelo de Alarcón, Spain; c.molinero.prof@ufv.es (C.M.); adrian.garrido@centta.es (A.G.-Z.); 2HUM-129 Behavior Analysis Group, Universidad de Granada, 18071 Granada, Spain

**Keywords:** Autism Spectrum Disorder, adolescents, social problem-solving skills, interpersonal skills, treatment, assessment, generalization

## Abstract

This study aims to examine the usefulness of an ad hoc worksheet for an Interpersonal Problem-Solving Skills Program (SCI-Labour) the effectiveness of which was tested by Bonete, Calero, and Fernández-Parra (2015). Data were taken from 44 adolescents and young adults with Autism Spectrum Disorder (ASD) (age M = 19.73; SD = 3.53; 39 men and 5 women; IQ M = 96.27, SD = 15.98), compared to a matched group (in age, sex, and nonverbal IQ) of 48 neurotypical participants. The task was conceived to promote the generalization of interpersonal problem-solving skills by thinking on different possible scenarios in the workplace after the training sessions. The results show lower scores in the worksheet delivered for homework (ESCI-Generalization Task) in the ASD Group compared to neurotypicals in total scores and all domains (Problem Definition, Quality of Causes, and Solution Suitability) prior to program participation. In addition, after treatment, improvement of the ASD Group was observed in the Total Score ESCI-Generalization Task and in the domains of Problem Definition, Quality of Causes, Number or Alternatives and Consequences, Time, and Solution Suitability. This is a valuable task in furthering learning within the SCI-Labour Program and may be a supplementary material in addressing the difficulties of interpersonal skills within this population, both in the workplace and in daily life. In conclusion, this task may provide useful information for identifying key difficulties among this population and could be implemented in a clinical setting as a complement to the SCI-Labour Program.

## 1. Introduction

Autism Spectrum Disorder (ASD) is a neurodevelopmental disorder characterized by deficits in social interaction and communication, restrictive patterns, and repetitive behaviors, interests or activities [1]. Numerous studies have found poor performance in social cognition and other skills that are involved in it (theory of mind, recognition of emotions, executive functioning, cognitive flexibility, and planning and inhibitory control [2,3,4,5,6]. These, along with verbal and nonverbal communication deficits, result in the lack of skills to deal with interpersonal conflicts, hindering social inclusion which increases as they reach adulthood [7,8,9].

There are many interventions for the development of socialization skills from early childhood to adulthood, some based on evidence-based practices [10,11,12]. There are at least three theoretical proposals in which social skills interventions could be classified: the *social skills to solve conflicts* approach [13] conceiving social skills as being domain-specific skills; *the social problem-solving process* [14,15], which considers that social problems are solved through a cognitive–emotional–behavioral process, and *interpersonal skills* framework as phases of the problem-solving process [16,17]. All these skills must be developed before adulthood, and in the absence of this development, psychological problems may arise [18].

During the course of training, changes are evaluated in a number of ways [19] using different assessment tools, such as questionnaires and self-report instruments, behavioral rating scales, lab-based behavioral observations, and performance tasks and expressive techniques [20]. Some problem-solving tasks are based on images, while others use comics, vignette sequences, etc. Working with ASD populations, different tasks have been used to assess performance and the improvement of social problem-solving skills [2,21,22,23,24,25,26,27,28,29]. However, problems tend to persist in maintaining these skills and with generalization to daily routines [30,31,32,33,34].

The goal of this study is to evaluate a performance task using a social problem-solving worksheet (ESCI-Generalization Task) applied during an intervention focused on interpersonal skills. Bonete et al. (2015) tested the ESCI-Labour Program effectiveness in a previous study. The theorical background of both the program and the task is based on the *interpersonal skills* approach [17,35], for which there is a lack of research in the ASD population [36]. The original study [23] programmed generalization based on the Train and Hope technique and Train Sufficient Examplers [31]. With this purpose, a mediational approach was used, and homework tasks were required. In this task, a social situation was described and a series of questions had to be answered. Each question was focused on a particular phase of the problem-solving process, the same phases participants were being trained with during the intervention. The first aim was to assess the validity of the ESCI-Generalization Task, discriminating between participants with ASD and neurotypicals. It was hypothesized that neurotypicals would score higher than the ASD Group prior to any training, and these differences would be smaller after the treatment. The second aim was to evaluate the potential utility of the ESCI-Generalization Task as an outcome measure of the effects of a manualized program for people with ASD in the context of an open clinical trial. We hypothesized that improvements in the ESCI-Generalization Task would be observed post-training, compared with pretreatment.

## 2. Materials and Methods

### 2.1. Participants

The sample was taken from data collected for a wider study examining the preliminary effectiveness of the Interpersonal Problem-Solving Skills for Workplace Adaptation [23]. The ASD Group was composed of 44 participants (39 men and 5 women) with ASD (ASD Group) aged between 16 and 30 years of age (M = 19.73; SD = 3.53) and with a global IQ within the limits of normality (M = 96.27; SD = 15.98) measured by the Reynolds Intellectual Screening Test [37]. The Comparison Group (CG) was the same as in the original study, recruited to match the ASD Group on sex, age, and nonverbal IQ [23]. It was composed of 48 subjects (42 men and 6 women) also aged 16 to 30 (M = 19.41; SD = 3.20) and with an IQ of M = 103.75; SD = 12.79. All participants were student volunteers with neurotypical development. All participants from the ASD Group had a confirmed diagnosis by gold standard measures [38,39] without any comorbidity or major psychiatric disorders, such as attention deficit hyperactivity disorder, obsessive compulsive disorder, or other disorders (See [23] for a full description of the sample). Intervention was implemented with the ASD Group exclusively who were asked to complete a worksheet after each session as homework. Only 37 participants from the ASD Group submitted the Pre ESCI-Generalization Task (completed after session 3), and 39 participants from the ASD Group submitted the Post ESCI-Generalization Task (completed after session 10). The CG only filled the Post ESCI-Generalization Task (session 10).

### 2.2. Intervention: SCI-Labour Program

The original SCI-Labour Program is a 10-week (90 min session once a week) interpersonal problem-solving training program [16] adapted for young people with ASD in the context of workplace adaptation. The SCI-Labour sessions were delivered in a small group format. A mediational approach was adopted through sequential training in a cognitive and metacognitive process; that is, each session developed one of the steps necessary to obtain the complete image of an interpersonal problem. Sessions 1 and 2 were introductory sessions: (I) introduction session and description of ASD characteristics and (II) conversational skills. The following sessions focused on a particular phase of the problem-solving process: (1) detecting and defining a social problem, (2) considering different perspectives, (3) looking for causes, (4) generating solutions, (5) considering consequences and (6) choosing the most adequate one, (7) making an action plan, (8) evaluating actions, and (9) facing failure. Each phase was based on exposure to typical interpersonal conflicts using social vignettes, videos, and scripts. These social situations were used to illustrate a problem and how it could be addressed using a cognitive process. Each session started with a personal problem (without social content), leading to a person-to-person problem, and to a group problem. A previous study confirmed the feasibility and effectiveness of using an open trial [23].

### 2.3. Target Measure: Social Problem-Solving Generalization Worksheet (ESCI-Generalization Task)

#### 2.3.1. Procedure

A performance task was designed to consolidate the skills developed in training with the SCI-Labour Program by practicing the content acquired after each session. The explicit aim was to promote generalization of the sequence of phases to solve an interpersonal problem by thinking of different possible scenarios in the workplace after training sessions. Over the course of the program, participants were asked to practice dealing with interpersonal conflicts at home using scripts [23]. This was an ad hoc homework task for the program, inspired by the material published by Paradiz [40]. Each worksheet describes an interpersonal conflict in a short story. The person must answer different questions following the sequence of steps of the social problem-solving process. Each question refers to a step which was addressed in a particular session. The worksheet template used for all the different scenarios is included in Appendix A.

The worksheet was introduced after the second session. In order to promote generalization, continuous practice was required [30,41]. Every week, participants were asked to analyze two scripts with different interpersonal conflicts: a *training task*, in which participants completed the cells of the worksheet referring to the social problem-solving skills trained during that session and a *generalization task* for which participants tried to complete the total sequence of steps for the adequate solution of that specific conflict. In the analysis, the generalization task after session 3 was used as the baseline (ESCI-Generalization Task Pre), while session 10 was taken as the Post-test to evaluate changes over time (ESCI-Generalization Task Post). Table 1 provides an overview of the scripts of social conflicts, session by session, used as generalization tasks of the program.

Different scenarios are described in each situation. However, the scripts for each session were selected from slightly different situations that were addressed during each treatment group session. In general, the worksheet was always the same: a short script followed by questions: (1) What clues do I use to detect there is a problem? (2) What is the problem? (3) What are the “main character’s” thoughts and feelings? (4) What do you think that the other person is thinking and feeling? (5) Point out as many possibilities as you can imagine, (6) List the different alternatives you can think as solutions, (7) Write at least one possible consequence that follows each alternative, and (8) Choose the solution that you think is most adequate to face this social situation.

In this study, only the task performance of scenarios Pre (session 3) and Post (session 10) were analyzed. These two situations were considered equivalent as they both addressed problems of shift work and how the decisions of others may affect performance at work.

#### 2.3.2. Response Coding

Each question of the ESCI-Generalization Task refers to a particular phase of the interpersonal problem-solving process (see Figure A1). Answers were coded into 10 categories: *Problem Definition* (*PD*), *Theory of Mind* (*ToM*), *Number of Causes* (*CAUS*), *Quality of Causes* (*CAUS-QLTY*), *Number of Alternatives* (*ALT*), *Quality of Alternatives* (*ALT-QLTY*), *Content of Alternatives* (*ALT-CONT*), *Number of Consequences* (*CONSQ*), *Time* (*T*), and *Solution Suitability* (*SS*). These items were based on definitions used in previous research examining interpersonal problem-solving skills [28,42,43,44]. A *Total score* was also calculated as the sum of the nine primary outcomes, except *Content of Alternatives* (ALT-CONT). This category, *Content of Alternatives* (ALT-CONT), was generated as a qualitative variable, exploring differences in the type of solutions generated, to see if the training also improved this aspect of social problem-solving skills. A description of each category rating is provided in Table 2.

An extra category was evaluated qualitatively, *Content of Alternatives* (ALT-CONT), according to 6 topics: (1) Search for help; (2) Verbal aggression; (3) No confrontation; (4) Compromise; (5) Negotiation/Agreement; and (6) Other, in order to analyze its frequency and if any changes appear after treatment.

#### 2.3.3. Coding Reliability

In coding the worksheets, the lead author (SB) trained two raters (blind study hypothesis) until interjudge reliability was established between them and the lead author in the coding of 20% of the total homework tasks. Checkers were considered in agreement when they gave the same rating to each category. Interrater reliability was calculated as ([Number of Agreement/Total Number of Codes] × 100). Reliability was considered acceptable once both raters achieved >80% of agreement in 28 aleatory selected worksheets. After that, all samples were coded by the same rater (blinded to group condition). The lead author did not rate any task.

### 2.4. Statistic Design

Data analyses were performed using the Statistical Package for the Social Sciences 22.0 (SPSS).

Given the features of the sample and the use of nominal variables, nonparametric measures were used. ESCI-Generalization Task validity was confirmed if the Total ESCI-Task discriminated between participants in the ASD Group and CG. Chi-squared tests were conducted comparing both groups in *Problem Definition* (PD), *Quality of Causes* (CAUS-QLTY), and *Solution Suitability* (SS) scores. The Mann–Whitney U test for independent samples was used to compare the ASD Group and CG scores in the main categories: ToM, CAUS, CAUS-QLTY, ALT, ALT-QLTY, CONSQ, T, and *Total Generalization task* score. Effect sizes were reported. As a complementary analysis, a logistic regression model was also calculated to evaluate predictive validity. The ESCI-*Total Generalization task* score was the independent variable (for the ASD Group the Post score was used), and the assigned group (ASD Group vs. CG) was the dependent variable. A qualitative analysis was presented when categorizing possible alternatives based on Content of Alternatives.

In order to examine the potential utility of the ESCI-Generalization Task to measure change after treatment, the Wilcoxon signed-rank test was used to compare pre- and post-treatment means in the ASD Group (*n* = 32 participants who complete Pre and Post) for the same categories.

## 3. Results

### 3.1. Differences between ASD Group Pre and Post-Treatment and Comparison Group

Examining the categorical variables, the comparison of the Pre-ASD Group and CG of the representative values of the contingency table and the differences between the groups is provided in Table 3. The CG showed fewer incorrect responses and more complete responses in Problem Definition (χ^2^ (2) = 17.41, *p* < 0.001), Quality of Causes (χ^2^ (2) = 27.96, *p* < 0.001), and Solution Suitability (χ^2^ (2) = 30.21, *p* < 0.001).

Comparing the Post-ASD Group and CG (Table 4), the CG showed fewer incorrect responses in Problem Definition (χ^2^ (2) = 16.38, *p* < 0.001) and more complete responses in Quality of Causes (χ^2^ (2) = 18.79, *p* < 0.001) and Solution Suitability (χ^2^ (2) = 8.32, *p* < 0.05). The comparison also revealed that a greater number of the Post-ASD Group participants offered completed responses than the Pre-ASD Group (comparing Table 3 and Table 4).

For the rest of the main variables, the Mann–Whitney U test showed significantly lower scores in the Pre-ASD Group in Number of Causes (*U* = 436.5, *z* = −4.07, *r* = 0.44), Number of Alternatives (*U* = 360, *z* = −4.76, *r* = 0.52), Quality of Alternatives (*U* = 248.5, *z* = −5.68, *r* = 0.62), Number of Consequences (*U* = 351.5, *z* = −4.83, *r* = 0.52), Time (*U* = 190.5, *z* = −6.26, *r* = 0.68), and Total ESCI-Generalization Task (*U* = 187, *z* = −6.22, *r* = 0.67) with a large effect size (see Table 5). No significant differences were found in ToM.

When examining the difference from the CG and Post-ASD Group, the Mann–Whitney U test showed significantly higher scores in the Post-ASD Group in the categories of Number of Consequences (*U* = 503.5, *z* = −3.75, *r* = 0.41), Time (*U* = 433, *z* = −4.36, *r* = 0.47), and in Total ESCI-Generalization Task with a large effect size (*U* = 355, *z* = −548, *r* = 0.59). The CG scored higher in Number of Alternatives (*U* = 576, *z* = −3.13, *r* = 0.32), Quality of Alternatives (*U* = 447, *z* = −4.18, *r* = 0.45), Number of Consequences (*U* = 503.5, *z* = −3.75, *r* = 0.41), Time (*U* = 433, *z* = −4.36, *r* = 0.47), and Total ESCI-Task (*U* = 355, *z* = −5.48, *r* = 0.59) (see Table 5).

Regarding effect size differences (comparing the performance of the Pre-ASD Group and CG against the Post-ASD Group and CG), it was observed that effect sizes decreased from large to medium in the variables Number of Alternatives, Number of Consequences, and Time according to Cohen’s Criteria (1988) (see Table 5). The Mann–Whitney U test showed significant differences between the ASD Group pre- and post-treatment scores and the CG.

Concerning the predictive validity of the ESCI-Generalization Task, the logistic regression model was statistically significant, χ^2^ (2, *N* = 92) = 51.53, *p* < 0.001. The model explained between 45.7% (Cox and Snell R square) and 61.3% (Nagelkerke R squared), generating an odds ratio of 1.08 for the Post-ASD Group. Thus, for every unit of increase in the *Total ESCI-Generalization Task*, the probability of having an ASD diagnosis was reduced by a factor of 1.08.

The qualitative analyses of the Content of Alternatives based on frequencies of answers showed that both the ASD Group and CG used mostly nonconfrontational and compromise actions.

### 3.2. ASD Group Differences before and after Treatment

Comparing Pre- and Postintervention outcomes in the ASD Group, Table 6 shows more correct responses in the categories Problem Definition (χ^2^ (2) = 32.20, *p* < 0.001), Quality of Causes (χ^2^ (2) = 7.85, *p* < 0.05), and Solution Suitability (SS) (χ^2^ (2) = 8.73, *p* < 0.05) after treatment.

The Wilcoxon signed-rank test showed significant increases after training in Number of Alternatives (*Z* = −2.15, *p* = 0.05, *r* = 0.47), Number of Consequences (*Z* = −2.07, *p* = 0.05, *r* = 0.23), Time (*Z* = −2.69, *p* = 0.01, *r* = 0.30), and *Total ESCI-Generalization Task* (*Z* = −2.00, *p* = 0.05, *r* = 0.10), although the effect size was small. There were no significant differences in Number of Causes and Quality of Alternatives (see Table 7). The Theory of Mind (ToM) category scored lower in Post (*Z* = −3.34, *p* = 0.01, and *r* = 0.38).

## 4. Discussion

In examining whether the *ESCI-Generalization Task* is useful for the evaluation of interpersonal conflict resolution skills among those with ASD, the aim was to determine if its application enhanced this learning process of interpersonal skills. This study also aimed to explore the effectiveness of this task in assessing changes after training and differences between neurotypicals and participants with ASD.

To ensure the validity of the *ESCI-Generalization Task*, raters were trained to reliably code.

The results of the analysis of the Pre-ASD Group and CG confirmed the hypothesis; higher scores were seen from the CG in all dimensions except for the ToM category [45,46]. These results indicate that for these variables, the *ESCI-Generalization Task* does effectively discriminate between the two groups and detects differences in the categories of the interpersonal problem-solving process.

Comparing the Post-ASD Group and CG, there was a decrease in the effect size (compared to the mean differences test with Pre-ASD and CG) in the variables of Number of Alternatives, Quality of Alternatives, Number of Consequences, and Time, indicating that the scores of the ASD Group, after intervention, are closer to those of the CG, as was the case with the scores for the different variables presented in the original study [23]. Significantly, there was an improvement not only in Number of Alternatives and Consequences but also in Quality of the Alternatives (based on the four aspects described) and the richness of the consequences, as there were improvements in the explanation of short-and long-term consequences. In general terms, one of the novelties of the *ESCI-Generalization Task* (as with other tasks of this type) is the quantification of the quality of the answers in aspects related to the kindness, efficacy, and relevance of the social responses to obtain an objective score.

With regards to post-treatment changes, the *ESCI-Generalization Task* proved to be a sensitive tool for measuring change after intervention. After training, the ASD Group showed great improvement in Problem Definition (71.8% of participants compared with 10.8% of participants in pretreatment) with enriched definitions. Although the Number of Causes showed nonsignificant changes, there was a clear improvement in Quality of Causes, 23.7% of participants included close and distant causes compared to 8.1% in pretreatment. Concerning the Number of Alternatives, the ASD Group significantly improved after training, even though the effect size was small. This is in line with the results of other studies in which generating solutions was one of the areas of change [30,47].

However, no improvements were found in the ASD Group after training in Quality of Alternatives; this category was composed of various aspects, such as Perspective of others, Activity, and Relevancy and Quality of Action, in which participants can show variability. In fact, standard deviation scores revealed that variability between subjects was as high as the mean scores. Individual changes were masked. For clinical purposes, computing the Reliable Change Index (RCI) [48] would be useful to observe individual changes.

As expected, in post-treatment, the ASD Group scored significantly higher in Number of Consequences and Time. This suggests that in addition to proposing a higher number of consequences in evaluating alternative solutions to a social problem, they learned to visualize to some extent the short-term and/or long-term consequences. In this regard, no studies were found that noted whether those with ASD distinguish between short- or long-term consequence when proposing solutions to interpersonal problems [21]. Looking at enrichment of the Solution Suitability chosen by participants, at post-treatment, 51.3% of the participants chose as the most appropriate action one with the higher score in the different aspects examined (*activity*, *relevance*, *perspective,* and *quality*) compared to only 18.9% at pretreatment. Finally, the Total ESCI-Generalization Task showed significant differences between the Pre-ASD and Post-ASD Groups, indicating a slight improvement. It could have positive ramifications on the use of the *ESCI-Generalization Task* as a measurement tool for outcomes with ASD sample groups. Contrary to expectations, for the Theory of Mind dimension, the average score of the ASD Group after training was lower than before, with no improvements in relation to the CG. Our interpretation is that the specific interpersonal situation described after session 10 (Post) might have presented a totally new challenge. Subjects had to try again to answer what thoughts or emotions the main character and the other person might have. Competence in Theory of Mind for the ASD population, as many studies have shown, may appear impaired or not [49]. A number of studies reported that ASD populations passed second-order tests of ToM successfully [4]. Our study was focused on young people, some of whom may have trained this precise ability during their childhood. This could explain the absence of changes and the similarities between groups. The original study showed similar upgrades in the rest of the outcome variables of the program [23]. It was expected that trained skills generalize to real-world life in the daily life. Programming generalization through exposure to different scenarios [30] in which social problem-solving skills play a role may have been mediating for improvements, although this variable was not controlled. Future studies could deepen in this controlling the possible practice effect of training with a structured task.

Among the limitations of the study is the size of the sample. Not all participants completed all the training sessions, nor did they all complete each assigned session task. This study represents only a preliminary step toward the validation of the *ESCI-Generalization Task*. It would have been interesting to analyze the rest of the worksheets (continuous probes of generalization) that participants completed after each session. Collecting maintenance data three months after the intervention would have enriched results consistency. Although the groups were homogeneous in verbal IQ, not all individuals with ASD have competent writing or reading comprehension skills, which may play a role in the general level of performance on tasks or specific variables. Someone unable to express the interpersonal situation in writing might find it very difficult to carry out the task. Finally, the complexity of the coding of tasks made the interpreting and correcting process more difficult. In this case, a double-blind coding was carried out with two trained people showing 80% agreement in their coding, but this is clearly limited and insufficient for extension to clinical use.

With regards to future lines of research, it may be interesting to explore if improvements in the ASD Group persist, that is, if a few years after the intervention they obtain similar competence when doing the worksheet with a different scenario. Studies have shown that although people with ASD can be taught tools to improve their social problem-solving skills, there is little evidence these improvements remain or can be generalized to other contexts [34,50,51,52]. Recent studies indicate that social skills training curricula are insufficient to improve the development of meaningful friendships among these individuals, interactional aspects of the program, and generalization tasks.

This study showed how the ASD Group improved its performance compared to the CG, but only the ASD Group that received training. Another study could assess the differences of learning achievement when both groups were compared in the training program. Another interesting possible line of research would be to validate the *ESCI-Generalization Task* in a wider neurotypical sample and to verify its cross-cultural validity. Finally, it would be interesting to obtain evidence about the effectiveness of this task, change its themes when applied to everyday situations and thus achieve better generalization skills, and to evaluate the use of this scheme and the acquired skills in the long term and under interactional conditions [53].

## 5. Conclusions

The *ESCI-Generalization Task* is a measurement tool for social problem-solving skills which, to some extent, show psychometric properties. The tool measures changes after treatment and distinguishes social problem-solving skills among those with ASD from neurotypicals. The results suggest that it is useful for detecting general changes among the sample, as progress is seen in solving interpersonally conflicting situations, particularly in the resolution phases of Problem Definition, Quality of Causes, Number or Alternatives and Consequences, Time, and Suitability of the Chosen Alternative. The *ESCI-Generalization Task* may also provide useful information in identifying key difficulties among this population in both working and everyday situations, generating an individualized profile for each person. In addition, this task can be implemented in the clinical field as a complement to the training program in the resolution of interpersonal problems and in order to further the learning of interpersonal skills and examining changes. This pilot study provides preliminary support for the *ESCI-Generalization Task* as part of a battery of assessment tools for various aspects of socialization.

## Figures and Tables

**Table 1 children-09-00166-t001:** Scripts of the ESCI-Generalization Task after each treatment session.

Variables	Task
Session 2	There was no generalization task for homework
Session 3 *	Carlos has left home late in the morning and has missed the bus, so he will arrive late to work. When he arrives to the office, he sees that his supervisor looks angry.
Session 4	Pedro has been working in a library for two weeks. The first days, the manager explained to him all the tasks that he had to do. Among them was to send the letters that the manager always left sealed on top of the table. Today, Pedro found five letters with the address written on them and prepared to send, but they are open. The manager had left, so Pedro decided to send them anyways. When the manager arrived and realized, he becomes very angry and told him off because the letters were for important people and they were incomplete, he shouldn’t have sent them. Pedro is very sad; he thinks that his boss has no reason to be angry like this.
Session 5	German works for a company, every employee works at their desk. Today, German takes a cup of coffee over to the boss’ desk. When he gives it to him, his hand trembles and the coffee falls onto his boss computer keyboard. The boss draws back abruptly, German can see the discomfort in his boss’ face.”
Session 6	Sonia works as a doorman for the cultural center for her neighbourhood. The manager of the cultural center has asked her to write up a document with the detailed timetable of the center’s activities. It took two days to finish it and she is very proud of how it looks with very pretty colors and writing. However, when she shows it to the manager, he tells her seriously that he doesn’t like how it’s done, and she will have to it all over.
Session 7	Julia works restocking a supermarket. Alongside her colleague, she makes sure all is done in the “Home” section. But her colleague, who has been working for the company longer than she has, most times isn’t very careful about placing the price labels, making the work slower and making it difficult for Julia to find what is missing.
Session 8	Felipe has been working as an electrician in a company for a short time. The boss askes him every day to stay a little longer after he finishes his shift. This is starting to become a problem for Felipe.
Session 9	Patricia works as a secretary. She has all documents filed in alphabetic order, but her boss doesn’t like how it’s done, and asks her to do it in a way that seems absurd to her.
Session 10 *	Jacinto is a security guard. He has finished his shift, but his supervisor, who is the one who must substitute him, hasn’t arrived.

*****: Coded and analyzed homework tasks.

**Table 2 children-09-00166-t002:** Description of dimensions of social problem-solving skills coded in the ESCI-Generalization Task.

Categories	Description
Problem Definition (PD)	Indicating if the problem was clearly stated (2 points), vaguely understood (1 point), or not understood at all (0 points). Maximum score: 2
Theory of Mind (ToM)	Score based on the understanding of emotions (1 point) and thoughts (1 point) about the principal actor and the other person involved. Maximum score: 4
Number of Causes (CAUS)	Number of causes attributed to the problem. 1 point was given for every plausible cause, relevant to the situation. Maximum score: 10
Quality of Causes (CAUS-QLTY)	The causes listed were categorized into “proximal” (refers to a cause with a recent effect) or “distant” (refers to a cause with a delayed effect). For coding, when a proximal and a distant cause are selected, the maximum score is given; if only a proximal or distant cause is selected, 1 point is given. Maximum score: 2
Number of Alternatives (ALT)	Participants were asked to list possible actions (plausible and relevant) for the principal actors to solve the scenario. Each plausible and relevant solution scores 1 point. Maximum score: 8
Quality of Alternatives (ALT-QLTY)	This score is the sum of four different subdomains exploring different aspects of the provided alternatives. A maximum of 8 for each of the 7 possible alternatives. Maximum score: 56 Activity (ACT): A solution is considered active if the main actor actually executes (2 points), but it is passive if action means to solve the problem through a third party not directly involved in the social problem (1 point). Relevancy (RELV): This scores if the action directly solves the issue (2 points) or is a step in a sequence of actions, indirectly solving the problem (1 point). Perspective (PERSP): 2 points if the participant took the other person involved into perspective and considered them affected by the action. Quality of Action (A-QLTY): 2 points when the action showed social sensitivity (coded in PERSP), and practical effectiveness (coded in RELV)
Number of Consequences (CONSQ)	Participants were required to list consequences to each alternative action that were plausible and relevant to the situation. 1 point for each option. Maximum score: 8
Time (T)	This task measured whether participants consider the duration of the consequence. This task was measured by whether it had short- (ST) or long-term (LG) consequences. 2 points for each option if both types of consequence were considered up to 8 consequences. Maximum score: 16
Solution Suitability (SS)	From the list of alternative actions, participants were to select the most appropriate and socially adequate actions regarding the situation. Maximum score: 2
Total ESCI-Generalization Task	With the sum of the responses of the subject in the previous dimensions, this task provides a total score. Maximum score: 108

**Table 3 children-09-00166-t003:** Contingency table of Chi Squared test in the categorical variables between Pre-ASD Group and CG.

Variables		Pre-ASD Group *N* (37)		CG *N* (48)		χ^2^	*p*	*r*
						(df = 2)		
		*N* (*%*)	*Res*	*N* (*%*)	*Res*			
PD	Incorrect	5 (13.5%)	2.8 *	0 (0%)	−2.8 *	17.41	0.000	0.45
	Partial	28 (75.7%)	4.9 *	25 (52.1%)	−4.9 *			
	Complete	4 (10.8%)	−7.8 *	23 (47.9%)	7.8 *			
CAUS-QLTY	Incorrect	12 (32.4%)	4.6 *	5 (10.4%)	−4.6 *	27.96	0.000	0.57
	Partial	22 (59.5%)	7.2 *	12 (25%)	−7.2 *			
	Complete	3 (8.1%)	−11.8 *	31 (64.6%)	11.8 *			
SS	Incorrect	23 (62.2%)	10.8 *	5 (10.4%)	−10.8 *	30.21	0.000	0.60
	Partial	7 (18.9%)	0.9	7 (14.6%)	−0.9			
	Complete	7 (18.9%)	−11.7 *	36 (75%)	11.7 *			

Note. PD: Problem Definition; CAUS-QLTY: Quality of Causes; SS: Solution Suitability; *N*: Number of participants; %: Percentage in groups; *Res*: Untyped waste; * (significant corrected residuals = −1.96 < 1.96); *p*: level of significance; χ^2^: Chi Squared; and *r*: Effect size.

**Table 4 children-09-00166-t004:** Contingency table of Chi Squared test in the categorical variables between Post-ASD Group and CG.

Variables		Post-ASD Group *N* (39)		CG *N* (48)		χ^2^	*p*	*r*
						(df = 2)		
		*N* (*%*)	*Res*	*N* (*%*)	*Res*			
PD	Incorrect	5 (12.8%)	2.8 *	0 (0%)	−2.8 *	16.38	0.000	0.43
	Partial	6 (15.4%)	−7.9 *	25 (52.1%)	7.9 *			
	Complete	28 (71.8%)	5.1 *	23 (47.9%)	−5.1 *			
CAUS-QLTY	Incorrect	18 (47.4%)	7.8*	5 (10.4%)	−7.8 *	18.79	0.000	0.46
	Partial	11 (28.9%)	0.8	12 (25%)	−0.8			
	Complete	9 (23.7%)	−8.7 *	31 (64.6%)	8.7 *			
SS	Incorrect	14 (35.9%)	5.5 *	5 (10.4%)	−5.5 *	8.32	0.016	0.31
	Partial	5 (12.8%)	−0.4	7 (14.6%)	0.4			
	Complete	20 (51.3%)	−5.1 *	36 (75%)	5.1 *			

Note. PD: Problem Definition; CAUS-QLTY: Quality of Causes; SS: Solution Suitability; *N*: Number of participants; %: Percentage in groups; *Res*: Untyped waste; * (significant corrected residuals = −1.96 < 1.96); *p*: level of significance; χ^2^: Chi Squared; and *r*: Effect size.

**Table 5 children-09-00166-t005:** Descriptive statistics and Mann–Whitney U test in the variables between the Pre-ASD Group (*n* = 37), Post-ASD Group (*n* = 39), and the Comparison Group (*n* = 48).

Variables	ASD Group *N* (37)			CG *N* (48)			*U*	*z*	*r*
	*Md*	*M*	*DT*	*Md*	*M*	*DT*			
ToM									
Pre	2	2.67	0.97	2	2.62	0.89	865.5	−0.22	0.02
Post	2	1.92	1.24	2			636.5 **	−2.83	0.31
Number of causes									
Pre	1	1.27	1.36	3	3.04	2.19	436.5 ***	−4.07	0.44
Post	1	1.02	1.11	3			389.5 ***	−4.75	0.51
Number of alternatives									
Pre	2	2.19	1.70	4	4.02	1.31	360 ***	−4.76	0.52
Post	3	2.92	1.69	4			576 **	−3.13	0.32
Quality of alternatives									
Pre	2	7.78	7.16	4	16.04	5.94	248.5 ***	−5.68	0.62
Post	10	9.51	6.85	15.5			447 ***	−4.18	0.45
Number of consequences									
Pre	7	2.10	1.95	15.5	1.51	1.51	351.5 ***	−4.83	0.52
Post	3	2.84	1.88	4			503.5 ***	−3.75	0.41
Time									
Pre	1	1.24	1.46	4	1.60	1.60	190.5 ***	−6.26	0.68
Post	2	2.28	1.99	4			433 ***	−4.36	0.47
Total ESCI-Task									
Pre	19	19.56	12.71	38.5	38.85	10.55	187 ***	−6.22	0.67
Post	23	21.27	14.50	38.5			355 ***	−5.48	0.59

Note. ***: *p* < 0.001; **: *p* < 0.01. *Md*: Median; *M*: Mean; *DT*: Typical deviation; *U*: Mann–Whitney U Statistic *N*: Number of participants; *z*: Normal distribution; and *r*: Effect size.

**Table 6 children-09-00166-t006:** Contingency table of Chi Squared test in the categorical variables between Pre- and Post- ASD Group.

Variables		Pre-ASD Group *N* (37)		Post-ASD Group *N* (39)		χ^2^	*p*	*r*
		*N* (*%*)	*Res*	*N* (*%*)	*Res*			
PD	Incorrect	5 (13.5%)	0.1	5 (12.8%)	−0.1	32.20	0.000	0.65
	Partial	28 (75.7%)	11.4 *	6 (15.4%)	−11.4 *			
	Correct	4 (10.8%)	−11.6 *	28 (71.8%)	11.6			
CAUS-QLTY	Incorrect	12 (32.4%)	−2.8 *	18 (47.4%)	2.8	7.85	0.020	0.32
	Partial	22 (59.5%)	5.7 *	11 (28.9%)	−5.7 *			
	Complete	3 (8.1%)	−2.9 *	9 (23.7%)	2.9 *			
SS	Incorrect	23 (62.2%)	5 *	14 (35.9%)	−5 *	8.73	0.013	0.34
	Partial	7 (18.9%)	1.2	5 (12.8%)	−1.2			
	Complete	7 (18.9%)	−6.1 *	20 (51.3%)	6.1 *			

Note. PD: Problem Definition; CAUS-QLTY: Quality of Causes; SS: Solution Suitability; *N*: Number of participants; %: Percentage in groups; *Res*: Untyped waste; * (significant corrected residuals = −1.96 < 1.96); *p*: level of significance; χ^2^: Chi Squared; and *r*: Effect size.

**Table 7 children-09-00166-t007:** Descriptive statistics of the quantitative variables and Wilcoxon signed-rank test between the Pre-ASD Group and the Post-ASD Group.

Categories	Pre-ASD Group *N* (32)			Post-ASD Group *N* (32)			*Z*	*r*
	*Md*	*M*	*DT*	*Md*	*M*	*DT*		
ToM	2	2.69	0.93	2	1.84	1.32	−3.34 **	0.38
Number of causes	1	1.21	1.43	1	1.09	1.17	−0.05	0.00
Number of alternatives	2	2.15	1.76	3	2.90	1.75	−2.15 *	0.24
Quality of alternatives	7	7.40	7.06	10	9.50	6.76	−1.53	0.17
Number of consequences	2	2.19	2.04	3	2. 90	2.01	−2.07 *	0.23
Time	1	1.22	1.47	2	2.41	2.08	−2.69 **	0.30
Total ESCI-Task	19	19.22	13.24	23	24.12	13.51	−2.00 *	0.10

Note. **: *p* < 0.01; *: *p* < 0.05. *Md*: Median; *M*: Mean; *DT*: Typical deviation; *N*: Number of participants; *Z*: Normal distribution; and *r*: Effect size.

## Data Availability

The data presented in this study are available on request from the corresponding author. The data are not publicly available due to data protection policy.

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
