# Peer review of "Generalization Task for Developing Social Problem-Solving Skills among Young People with Autism Spectrum Disorder"

_children, 2022, doi:10.3390/children9020166_

Round 1

Reviewer 1 Report

Main message of the article

The study by Bonete et al. aimed to examine the usefulness of the ESCI-Generalization Task when used in an Interpersonal Problem-Solving Skills program for individuals with Autism Spectrum Disorder (ASD). The results suggests that the ESCI-Generalization Task allows to distinguish between adolescents and young adults with ASD and a control group composed by neurotypical matched individuals. Also, the authors claimed that the ESCI-Generalization Task allows to further the learning of social problem-solving skills within the SCI-Labour Program.

General Judgment Comments

The article is clearly written, especially the parts regarding the SCI-Labour Program and the Target measure. The study methodology is appropriate. The title and the abstract do not refer correctly to the sample population and should highlight that the study was conducted with adolescents and young adults with ASD. The major weakness of the study regards the data analysis, which is not appropriate, and its results are in part misinterpreted. Also, the sample sizes are small with a strong prevalence of male participants. Throughout the manuscript, results need to be reported not only in tables, but also in the text following the appropriate standard.

I would recommend the article to undergo Major Revision because it requires more work on the statistical analysis and on the interpretation of the results.

Major Issues

  • The sample of participants mostly represents the male population with ASD and the does not reflect the ASD gender distribution that is commonly found in the scientific literature. Thus, the study results are scarcely generalizable to the female population with ASD. Authors should highlight this limitation in the Discussion of their results.
  • Scarce details are provided in terms of the diagnosis of the ASD group, and this could potentially limit the generalizability of the study’s results. In the ASD group, were there people with intellectual disability or with compromission of language? What was the severeness of their ASD? Knowing these details would allow to understand whether the results are generalizable to the whole group of individuals with ASD, or only to the “high-functioning” ones.
  • Why did the authors match the control and the ASD groups only in terms of verbal IQ and not general IQ? Please clarify in the manuscript. Also, please provide details on the verbal IQ within the ASD group.
  • Materials & Methods, line 79-81: please provide information on the gender, age, IQ distribution within the 37 participants in the ASD groups that filled in the Pre ESCI-Generalization task and the 39 that submitted the Post ESCI-Generalization task. This would help on understanding the representativeness of the groups and to understand the appropriateness of the control group.
  • In the manuscript, the authors stated that only 37 participants with ASD completed the Pre ESCI-Generalization task and only 39 completed the Post ESCI-Generalization task. It is not clear what the authors did with the data from participants that did not complete either one of the tasks. Since the study aims to assess any potential improvement in performances on social problem-solving after the training, the authors need to exclude from the study all data coming from participants who did not complete one of the two ESCI-Generalization tasks (either Pre or Post). In fact, to use the Wilcoxon signed-rank test, each participant needs to have data on both the Pre and Post ESCI-Generalization tasks, otherwise the results of the test are misleading. Hence, the subsection “3.2. ASD Group differences before and after treatment” needs to be revised, together with the interpretation of its results throughout the manuscript. Also, the number of completed worksheets should be controlled in the study when comparing data from Pre and Post ESCI-Generalization tasks on the ASD group.
  • Materials & Methods, lines 173-174: the authors stated that “Chi-squared tests were conducted comparing both groups in Problem Definition (PD), Quality of Causes (CAUS-QLTY) and Solution Suitability (SS) scores”. Why were only these three domains of the ESCI-Generalization task selected to be tested among the others using Chi-Squared test? Why were they not tested also by using Mann Whitney U-test as the remaining variables? Also, was any statistical correction adopted to adjust the significance threshold and control for multiple comparisons?

Minor Issues

  • Title: To have a clearer picture of the sample of participants in the current research, I would suggest to replace “young people” with “adolescents and young adults” in the title.
  • In the keywords, to better represent the sample of participants in the study, I would suggest either to replace “adolescents” with “adolescents and young adults” or to add “young adults” to the current keywords.
  • Abstract, lines 9: replace “adolescents” with “adolescents and young adults”.
  • Abstract, lines 10: The authors stated “[…] compared to a matched group (in age, sex, IQ) of 48 neurotypical participants”. In terms of which IQ was the samples matched? General, verbal, non-verbal, others?
  • Abstract, line 11: The authors stated that “The task was conceived to promote generalization”. Please clarify the context to which the skill was aimed to be generalized.
  • Abstract, line 11: what is the ESCI-Generalization Task? Please clarify for the readers.
  • Introduction, lines 25-27: the sentence “Autism Spectrum Disorder (ASD) is a neurodevelopmental disorder characterized by deficits in social interaction and communication, restrictive patterns and repetitive behaviors, interests or activities” requires a reference, probably the best one would be the DSM-V reference.
  • Introduction, lines 27-29: this sentence needs to be justified by articles not only focusing on high functioning autism [1] or by the comorbidity of attention deficits in autistic individuals [2], but by references that potentially explore all the domains that the authors claim be “weaker” in individuals with ASD. The use of meta-analysis or literature reviews is recommended to justify the statement.
  • Introduction, line 35: what do the authors mean by “conceiving social skills as being discrete skills”? In this context, is “discrete” a synonym of “domain-specific”? Please clarify.
  • Introduction, line 46: “However, problems tend to persist in maintaining these skills and with generalization”. Please clarify the context to which the skill was aimed to be generalized.
  • Introduction, lines 49-50: the authors stated that “The theorical background of both, the program and the task is based on the interpersonal skills approach [13]”, why was this approach chosen over the other two? Please clarify the benefits of adopting the interpersonal skills approach in the manuscript.
  • Introduction, line 50: “The original study [19] programmed […]”, which original study are the authors referring to? In what way are the original and current studies related? Please clarify.
  • Materials & Methods, lines 66-67: to help understanding the generalizability of the current results to the population of individuals having ASD, please report also the range of values for the measured IQs.
  • Materials & Methods, lines 66-67: “Tarjet” to “Target”?
  • Materials & Methods, line 106: the authors stated “The explicit aim was to promote generalization”. Please clarify the context to which the skill was aimed to be generalized.
  • Materials & Methods, lines 148-149: the authors stated that “A Total score was also calculated as the sum of the nine primary outcomes, except ALT-CONT”. Why ALT-CONT was not included in the computation of the total score?
  • In Tables, please do not repeat the abbreviation in the Note of the table. “p:” is repeated two times in Table 3 and Table 4.
  • Results, lines 196-199: “When comparing Pre and Post in ASD-Group, it was clearly observed that ASD Group showed more complete responses and fewer incorrect responses concerning the Problem Definition, the Quality of the possible causes and the Solution Suitability (see Table 4)”. Table 4 does not show the comparison between Pre and Post ESCI-Generalization scores in ASD group, but the comparison between Post ASD group and CG. Also, any comparison in terms of improvement/decrease between Pre and Post ESCI-Generalization scores in ASD group is not valid in the current paper, since the scores of the Pre and Post tasks were not collected on the same participants.
  • Results, lines 220-223: “Regarding effect sizes differences (by comparing the performance of the Pre-ASD group and CG against the Post-ASD group and CG), it was observed that effect sizes decrease from long to medium respectively in the variables Number of Alternatives, Number of Consequences and Time according to Cohen's Criteria (1988) (see Tables 5 and 6)”. Why did the authors reference Table 6?
  • Tables 3, 4, and 6’s titles need to more explanatory. Please revise them.
  • Discussion, line 321: in the limitations, authors need to highlight the fact that their ASD group was almost uniquely composed by males.
  • Please, in the Results section, report the results in the appropriate format also within the main text, not only in the Tables.

Author Response

Q1-English

Grammar, spelling and punctuation errors were corrected. A native translator reviewed the full article

Q2-. The title and the abstract do not refer correctly to the sample population and should highlight that the study was conducted with adolescents and young adults with ASD

Initially it was identified as “adolescents and adults with ASD”, but we changed it in order to shorten the title. We believe that “young people” encompasses all participants from 16 to 30 years old, especially when medium age is 19 years old.

Q3- The data analysis are not appropriate and results are in part misinterpreted.

-          Report the results in tables and text

Data analysis are justified and results reviewed as reviewer required.

-          Results were reported in tables and text.

Q4- The sample sizes are small and a strong prevalence of male participants.

-          Author should mention it as a limitation of the results

This study is a second aim of a broader project. Program effectiveness was first published (Bonete et al., 2015) in which samples are fully described.

 The ASD sample size is wide enough to explore the program effectiveness (other studies explore effectiveness with smaller samples such as:

-          Mpella, M., Evaggelinou, C., Koidou, E., & Tsigilis, N. (2019). The Effects of a Theatrical Play Programme on Social Skills Development for Young Children with Autism Spectrum Disorders. International Journal of Special Education33(4), 828-845.

-          Ridderinkhof, A., de Bruin, E. I., Blom, R., & Bögels, S. M. (2018). Mindfulness-based program for children with autism spectrum disorder and their parents: direct and long-term improvements. Mindfulness9(3), 773-791.)

It is also known that ASD prevalence of diagnosis is higher in males:

-          Anderson, A. H., Stephenson, J., Carter, M., & Carlon, S. (2019). A systematic literature review of empirical research on postsecondary students with autism spectrum disorder. Journal of Autism and Developmental Disorders49(4), 1531-1558.

-          French, L., & Kennedy, E. M. (2018). Annual Research Review: Early intervention for infants and young children with, or at‐risk of, autism spectrum disorder: a systematic review. Journal of Child Psychology and psychiatry59(4), 444-456.

We prefer not to eliminate females from the data as we are exploring social cognition improvements on a task used during the program, and the original ASD Group sample (n=50) was equivalent in age, IQ and sex to a CG (n=50). But, as it is suggested, we point out this issue in limitations section

Q5- Scarce details are provided in terms of the diagnosis of the ASD Group

-          Were there people with intellectual disability? With compromission of language? In order to know if the results could be generalized the to the whole group of individuals with ASD or only “high-functioning”

-          Why did the authors match the control and the ASD groups only in terms of verbal IQ and not general IQ? Please clarify in the manuscript. Also, please provide details on the verbal IQ within the ASD group.

-          please provide information on the gender, age, IQ distribution within the 37 participants in the ASD groups that filled in the Pre ESCI-Generalization task and the 39 that submitted the Post ESCI-Generalization task. This would help on understanding the representativeness of the groups and to understand the appropriateness of the control group.

Description of diagnosis was brief as it is widely described in the original study (in which program effectiveness was explored): Bonete, Calero & Fernández-Parra, 2015. This was now clarified in abstract, introduction section and participants section in order to avoid being redundant.

-          Participants were high-functioning ASD participants (IQ around normality as it was specified). As there is a wide heterogeneity among people with ASD, we matched the group of comparison so that variability would not be due to these variables.

-          We reported sex, age and IQ of the full sample of 44 participants (ASD Group) that fill the worksheet that is analyzed in the article. There were 6 more participants who participated in the program that did not deliver any worksheet, so they were considered lost data. There were also 2 CG participants that did not deliver the worksheet so had to be remove from the analysis, that is why CG=N=48.

-           

-          Part of the analysis could be done even though some participants only deliver PRE or POST worksheet but, analysis are group based so we prefer not to loose any more data.   

Q6- In the manuscript, the authors stated that only 37 participants with ASD completed the Pre ESCI-Generalization task and only 39 completed the Post ESCI-Generalization task. It is not clear what the authors did with the data from participants that did not complete either one of the tasks. Since the study aims to assess any potential improvement in performances on social problem-solving after the training, the authors need to exclude from the study all data coming from participants who did not complete one of the two ESCI-Generalization tasks (either Pre or Post). In fact, to use the Wilcoxon signed-rank test, each participant needs to have data on both the Pre and Post ESCI-Generalization tasks, otherwise the results of the test are misleading. Hence, the subsection “3.2. ASD Group differences before and after treatment” needs to be revised, together with the interpretation of its results throughout the manuscript. Also, the number of completed worksheets should be controlled in the study when comparing data from Pre and Post ESCI-Generalization tasks on the ASD group.

As requested, the Wilcoxon signed-rank test, on subsection 3.2, has been recalculated with the 32 participants that had both, Pre and Post tests; interpretation of the results were also reviewed through all the paper.

Q7- Materials & Methods, lines 173-174: the authors stated that “Chi-squared tests were conducted comparing both groups in Problem Definition (PD), Quality of Causes (CAUS-QLTY) and Solution Suitability (SS) scores”. Why were only these three domains of the ESCI-Generalization task selected to be tested among the others using Chi-Squared test? Why were they not tested also by using Mann Whitney U-test as the remaining variables? Also, was any statistical correction adopted to adjust the significance threshold and control for multiple comparisons?

This variables were analysed as categorical variables due to its short range of scores: 2 points COMPLETE, one point PARTIALLY COMPLETED, 0 points WRONG.

Q8- I would suggest either to replace “adolescents” with “adolescents and young adults” or to add “young adults” to the current keywords.

We prefer not to change it due to the scope of the journal.

Q9- Abstract, lines 9: replace “adolescents” with “adolescents and young adults”.

Done

Q10- Abstract, lines 10: The authors stated “[…] compared to a matched group (in age, sex, IQ) of 48 neurotypical participants”. In terms of which IQ was the samples matched? General, verbal, non-verbal, others?

Due to ASD people heterogeneity, Non-verbal IQ is considered more representative. It is now explicitly written through the text (as it was already in the original study). Even though all of participants were under the limits of normality, verbal skills could be altered (as part of the diagnosis). This is an issue that is risen in the limitations section.

Q11- Abstract, line 11: The authors stated that “The task was conceived to promote generalization”. Please clarify the context to which the skill was aimed to be generalized.

To promote generalization by thinking on different possible scenarios in the place of work after training sessions.

Q12- Abstract, line 11: what is the ESCI-Generalization Task? Please clarify for the readers.

Done

Q13- Introduction, lines 25-27: when describing ASD, add DSM-V reference

Done

Q14- Introduction, lines 27-29: this sentence needs to be justified by articles not only focusing on high functioning autism [1] or by the comorbidity of attention deficits in autistic individuals [2], but by references that potentially explore all the domains that the authors claim be “weaker” in individuals with ASD. The use of meta-analysis or literature reviews is recommended to justify the statement.

Done

Q15- Introduction, line 35: what do the authors mean by “conceiving social skills as being discrete skills”? In this context, is “discrete” a synonym of “domain-specific”? Please clarify.

Yes, that is right. We change the term.

Q16- Introduction, line 46: “However, problems tend to persist in maintaining these skills and with generalization”. Please clarify the context to which the skill was aimed to be generalized.

It is frequently expected that the learning generalizes to the daily routine when interpersonal problems rise. And it is frequently observed that even though participants are doing well during the intervention sessions they do not seem to link with their personal situation. We include some more references.

Q17- Introduction, lines 49-50: the authors stated that “The theorical background of both, the program and the task is based on the interpersonal skills approach [13]”, why was this approach chosen over the other two? Please clarify the benefits of adopting the interpersonal skills approach in the manuscript.

Justification was included in the text with some references.

Q18- Introduction, line 50: “The original study [19] programmed […]”, which original study are the authors referring to? In what way are the original and current studies related? Please clarify.

This study is part of a broader research, what we called the original study in which program effectiveness was tested. From that project, now we analyze the changes that could be observed on the worksheet.

Q19- Materials & Methods, lines 66-67: to help understanding the generalizability of the current results to the population of individuals having ASD, please report also the range of values for the measured IQs.

Q20- Materials & Methods, lines 66-67: “Tarjet” to “Target”?

-          Materials & Methods, line 106: the authors stated “The explicit aim was to promote generalization”. Please clarify the context to which the skill was aimed to be generalized.

Spelling mistake was changed.

Clarification is made in lines 119-120.

Q21- Materials & Methods, lines 148-149: the authors stated that “A Total score was also calculated as the sum of the nine primary outcomes, except ALT-CONT”. Why ALT-CONT was not included in the computation of the total score?

ALT-QLTY refers to quality of alternatives, the type of alternatives that participant generates. Topics are describe in lines 172-174. This is a qualitative classification therefore it can not be summed with the rest of the variables. It has been explicitly described now (line 165)

Q22- In Tables, please do not repeat the abbreviation in the Note of the table. “p:” is repeated two times in Table 3 and Table 4.

Corrected

Q23- Results, lines 196-199: “When comparing Pre and Post in ASD-Group, it was clearly observed that ASD Group showed more complete responses and fewer incorrect responses concerning the Problem Definition, the Quality of the possible causes and the Solution Suitability (see Table 4)”. Table 4 does not show the comparison between Pre and Post ESCI-Generalization scores in ASD group, but the comparison between Post ASD group and CG. Also, any comparison in terms of improvement/decrease between Pre and Post ESCI-Generalization scores in ASD group is not valid in the current paper, since the scores of the Pre and Post tasks were not collected on the same participants.

Writing was confusing. We change it to be clearer (lines 216-218)

The comparison is made on the same participants.

Q24- Results, lines 220-223: “Regarding effect sizes differences (by comparing the performance of the Pre-ASD group and CG against the Post-ASD group and CG), it was observed that effect sizes decrease from long to medium respectively in the variables Number of Alternatives, Number of Consequences and Time according to Cohen's Criteria (1988) (see Tables 5 and 6)”. Why did the authors reference Table 6?

It was corrected

Q25- Tables 3, 4, and 6’s titles need to more explanatory. Please revise them.

They were reviewed and changed.

Q26- Discussion, line 321: in the limitations, authors need to highlight the fact that their ASD group was almost uniquely composed by males.

Done

Q27- Please, in the Results section, report the results in the appropriate format also within the main text, not only in the Tables.

Done.

Reviewer 2 Report

Re: Children-1536394 Bonete et al report on an assessment task for social problem solving in ASD. It is of interest but there are several issues that would need to be addressed.

In the abstract, not sure if it is needed to add ‘detecting effects of’ before ‘an Interpersonal Problem-Solving Skills program’ in the first line for clarity. See the issue raised below in the Discussion.

Introduction- there was one particular odd point made, after ‘have found social cognition deficit, specifically,…’ the authors then listed several things that were social cognition related and other things unrelated to social cognition.

Also, there were a number of grammatical errors. In the phrase above, ‘cognition deficit’ should be ‘cognition deficits’ (line 27).

Line 42, a comma is needed after ‘behavioral observation’,

line 43 change ‘other’ to ‘others’ and add a comma after ‘vignettes’,

line 49 add comma after ‘task’,

line 54 insert ‘the’ before ‘same phases’.

Methods- please specify which gold-standard measures were used to confirm diagnosis for the reader. Also, ‘(90 min sessions)’- is that once weekly?

Figure 1 is much too small to read the contents. This needs to be enlarged.

In Table 2- under ALT-QLTY- please be clear for the reader how the number 56 is reached- presumably a maximum of 7 for the subdomains for each of up to 8 alternatives.

For CONSQ clarify that it is 1 per consequence, as was made clear for previous measures, and for Time, clarify the scoring as well. The Coding reliability was confusing- two raters were trained, but was only one rater used in the entire study? That is how it is stated. Also, need to specify if the raters were blinded to group and pre/post.

Also, there are grammar issues.

Line 68 change ‘matched’ to ‘match’,

line 69 change ‘by’ to ‘of’,

line 77 change ‘fulfil’ to ‘fill’,

line 79 insert ‘the’ before ‘Pre’,

line 80 insert ‘the’ before ASD and ‘the’ before Post, and line 81 change ‘fulfilled’ to ‘filled’, and line 90 change ‘[‘ to ‘(‘. Results- in the footnotes of Table 3 the χ is missing for χ2.

Line 220 – do the authors mean ‘large’ rather than ‘long’?

Line 221 the call out for Tables 5 and 6 seems wrong. Table 5 is correct but there is no Number of Alternatives, Number of Consequences, or Time in Table 6.

Line 225 the symbol is also missing. There are also grammatical issues here as well.

Line 218 change ‘sizes’ to ‘size’,

line 222 fix ‘U-Mann Whitney’, and insert ‘the’ before ‘ASD,

line 227 change ‘odd’ to ‘odds’ and insert ‘the’ before ‘Post’,

line 241 change ‘increasement’ to ‘increase’, and line 245 change ‘in’ to ‘at’.

Discussion- On line 252-3 the authors say ‘to determine if its application enhanced this learning process of interpersonal skills’ but on line 275-6 the authors say ‘proved to be a sensitive tool in measuring change’- which is it? An intervention or a measurement tool? This affects the Abstract comment as above. This needs to be clearer throughout the paper- is this just a measurement during a broader intervention? Or an intervention itself? Also, line 256- are the rater blinded as well as trained? Would rephrase the ‘indicating a slight improvement’ as it was not significant- would be better to just say that the change did not reach significance.

The discussion of the Theory of Mind in ASD should be expanded and more thoughtful in lines 309-313, reflecting an understanding of its salience in ASD. The points made in lines 314-317 and 334-336 are important- it is not clear how well this might generalize. The authors should discuss this more thoughtfully. This may simply reflect a practice effect on a structured task. This possibility should be recognized. On line 326, do the authors mean to add ‘performance on’ before ‘tasks’? Also, the sentence spanning lines 326-328 does not make sense- please fix. It does not become stated until line 329 that the coding was double-blind- this needs to be clear earlier in the paper, if this is the case.

There are also grammar issues here.

Line 276 change ‘in’ to ‘for’,

line 280 change ‘closed’ to ‘close’,

line 297 add a comma after ‘action’,

line 309 add ‘the’ before ‘ASD’,

line 337 add ‘the’ before ‘ASD’ and change ‘shorten’ to ‘shortens’, and line 338 add ‘the’ before ‘ASD’. Conclusions- change ‘measure’ to ‘measurement’.

Author Response

Thank you very much for all the comments. 

Q1- Moderate English changes required

- English was reviewed by a native translator

Q2- In the abstract, not sure if it is needed to add ‘detecting effects of’ before ‘an Interpersonal Problem-Solving Skills program’ in the first line for clarity. See the issue raised below in the Discussion.

Q3- Introduction- there was one particular odd point made, after ‘have found social cognition deficit, specifically,’ the authors then listed several things that were social cognition related and other things unrelated to social cognition.

It exists some evidence that the areas that are mentioned (theory of mind, recognition of emotion, executive functioning, cognitive flexibility, planning and inhibitory control) are related some how to social cognition

Q4- Also, there were a number of grammatical errors.

-          In the phrase above, ‘cognition deficit’ should be ‘cognition deficits’ (line 27).

-          Line 42, a comma is needed after ‘behavioral observation’,

-          line 43 change ‘other’ to ‘others’ and add a comma after ‘vignettes’,

-          line 49 add comma after ‘task’,

-          line 54 insert ‘the’ before ‘same phases’.

Corrected

Methods- please specify which gold-standard measures were used to confirm diagnosis for the reader. Also, ‘(90 min sessions)’- is that once weekly?

Gold-standard measures are ADI-R and ADOS- Both worldwide known in the assessment of autism, they are described in the original study that is mention (Bonete et al., 2015). We believe it is not the focus of this article. We ask the reader to check for the previous study.

-          90 min once a week

Q5- Figure 1 is much too small to read the contents. This needs to be enlarged.

It was finally changed to Appendix section so that it could be enlarged.

Q6- In Table 2- under ALT-QLTY- please be clear for the reader how the number 56 is reached- presumably a maximum of 7 for the subdomains for each of up to 8 alternatives.

Thank you for the observation: a maximum of 8 for each option up to 6 alternatives that could be provided in the template worksheet.

Q7- For CONSQ clarify that it is 1 per consequence, as was made clear for previous measures, and for Time, clarify the scoring as well.

Done

Q8- The Coding reliability was confusing- two raters were trained, but was only one rater used in the entire study? That is how it is stated. Also, need to specify if the raters were blinded to group and pre/post.

It is right.

It was done.

Q9- Also, there are grammar issues.

Line 68 change ‘matched’ to ‘match’,

line 69 change ‘by’ to ‘of’,

line 77 change ‘fulfil’ to ‘fill’,

line 79 insert ‘the’ before ‘Pre’,

line 80 insert ‘the’ before ASD and ‘the’ before Post, and line 81 change ‘fulfilled’ to ‘filled’, and line 90 change ‘[‘ to ‘(‘.

Results- in the footnotes of Table 3 the χ is missing for χ2.

Line 220 – do the authors mean ‘large’ rather than ‘long’?

Line 221 the call out for Tables 5 and 6 seems wrong. Table 5 is correct but there is no Number of Alternatives, Number of Consequences, or Time in Table 6.

Line 225 the symbol is also missing. There are also grammatical issues here as well.

Line 218 change ‘sizes’ to ‘size’,

line 222 fix ‘U-Mann Whitney’, and insert ‘the’ before ‘ASD,

line 227 change ‘odd’ to ‘odds’ and insert ‘the’ before ‘Post’,

line 241 change ‘increasement’ to ‘increase’, and line 245 change ‘in’ to ‘at’.

Done

Round 2

Reviewer 1 Report

the article is now acceptable

Author Response

Thank you very much for all your comments. The manuscript has improved thanks to your revision.  Sincerely,

Reviewer 2 Report

Re: Children-1536394R1

Bonete et al have resubmitted their report on an assessment task for social problem solving in ASD.  It is of interest several issues remain that would need to be addressed.  It seems that the authors have corrected my editorial comments but have skipped most of my more substantive comments.

Introduction- there was one particular odd point made, after ‘have found social cognition deficit, specifically,…’ the authors then listed several things that were social cognition related and other things unrelated to social cognition.  This was raised at the previous review and is not addressed.

Methods- For ‘(90 min sessions)’- is that once weekly?  Figure 1 is much too small to read the contents.  This needs to be enlarged.  The Coding reliability was confusing- two raters were trained, but was only one rater used in the entire study?  That is how it is stated.  Also, need to specify if the raters were blinded to group and pre/post.  These were raised at the previous review and not addressed.

Results- line 243 fix ‘U-Mann Whitney’.

Discussion- On line 277-8 the authors say ‘to determine if its application enhanced this learning process of interpersonal skills’ but on line 300-1 the authors say ‘proved to be a sensitive tool for measuring change’- which is it? An intervention or a measurement tool?  This needs to be clearer throughout the paper- is this just a measurement during a broader intervention? Or an intervention itself?  Would rephrase the ‘indicating a slight improvement’ as it was not significant- would be better to just say that the change did not reach significance.  The discussion of the Theory of Mind in ASD should be expanded and more thoughtful in lines 332-6, reflecting an understanding of its salience in ASD.  The points made in lines 338-341 and 358-363 are important- it is not clear how well this might generalize.  The authors have begun to discuss this but should acknowledge issues about how this may in part be driven by a practice effect on a structured task.  This possibility should be recognized.  On line 350, do the authors mean to add ‘performance on’ before ‘tasks’?  Also, the sentence spanning lines 350-2 does not make sense- please fix.  These issues were raised previously and some are partially addressed, but most are not addressed.

Conclusions- change ‘measure’ to ‘measurement’ in the first sentence. This was raised at the previous review and is not addressed.

Author Response

My co-authors and I are pleased with the feedback we have received on our manuscript. We think the paper is greatly improved, based on the changes we made in response to your suggestions. Below I summarize each point you raised. In the second column, I summarize how the point/concern was addressed. We hope we answered all your observations. Thank you very much,

Reviewer comments

Response

Introduction- there was one particular odd point made, after ‘have found social cognition deficit, specifically,’ the authors then listed several things that were social cognition related and other things unrelated to social cognition.  This was raised at the previous review and is not addressed.

We change the sentence to be more accurate. There are social cognition deficits AND other skills that are involved (some of them are related to socialization and others do not have a direct relation but seem to be necessary for the development of socialization skills. Reported references point out to each of these domains and its relation with social cognition and symptoms in ASD population:

- Theory of mind, recognition of emotions and executive functioning were studied in Stichter et al., 2010

Theory of mind was developed in Bowler, 1992 (as one of the multiple references concerning this area)

Recognition of emotions is studied in Kennedy & Adolphs, 2012

Executive functioning is further mention in Robinson et al., 2009

Cognitive Flexibility, planning and inhibitory control was addressed in  Berenguer et al., 2015 

Methods- For ‘(90 min sessions)’- is that once weekly?  Figure 1 is much too small to read the contents.  This needs to be enlarged.  The Coding reliability was confusing- two raters were trained, but was only one rater used in the entire study?  That is how it is stated.  Also, need to specify if the raters were blinded to group and pre/post.  These were raised at the previous review and not addressed.

Line 94 specifies “once a week”

Figure 1 has been removed and sent to appendix 1, in order to enlarge the image.

Yes, you are right, the coding reliability is as stated (lines 170 to 177). At first, both raters coded 20% of the tasks, then, when reliability was achieved, only one of them coded the rest of the tasks. The 100% of the data used for the analysis was always from the same rater.

Finally, raters were blinded to the group; not to the pre/post. This information is in the manuscript (line 175).

Results- line 243 fix ‘U-Mann Whitney’.

Corrected

Discussion- On line 277-8 the authors say ‘to determine if its application enhanced this learning process of interpersonal skills’ but on line 300-1 the authors say ‘proved to be a sensitive tool for measuring change’- which is it? An intervention or a measurement tool?  This needs to be clearer throughout the paper- is this just a measurement during a broader intervention? Or an intervention itself? 

Would rephrase the ‘indicating a slight improvement’ as it was not significant- would be better to just say that the change did not reach significance. 

The discussion of the Theory of Mind in ASD should be expanded and more thoughtful in lines 332-6, reflecting an understanding of its salience in ASD. 

The points made in lines 338-341 and 358-363 are important- it is not clear how well this might generalize.  The authors have begun to discuss this but should acknowledge issues about how this may in part be driven by a practice effect on a structured task.  This possibility should be recognized. 

On line 350, do the authors mean to add ‘performance on’ before ‘tasks’?  Also, the sentence spanning lines 350-2 does not make sense- please fix.  These issues were raised previously and some are partially addressed, but most are not addressed.

ESCI-Generalization task includes different scripts that were used one per session during a broader intervention. The Program (SCI-Labour) consisted of 90 min 10 weekly sessions. In order to analyze changes after the intervention, we coded participants performance for the first and the last script of the ESCI- Generalization task. Since it has been sensitive to changes, in the discussion we proposed it can be used as a measurement tool.

When sample numbers were matched, and the analyses were repeated for the Wilcoxon test (suggested by the other reviewer), this difference appeared significant.

We tried to clarify our intention to rise this issue (line 335) and added two references concerning Theory of Mind and its importance in ASD people. We consider we should not extend much longer in this area but going through all the dimensions that covers the task.

We agree with the observation concerning the possibility of practice effect on a structured task, we added the idea (line 340)

We simplified the idea in lines 350-51. We hope it is clearer now.

Conclusions- change ‘measure’ to ‘measurement’ in the first sentence. This was raised at the previous review and is not addressed.

We apologize for skipping this correction when reviewing the rest of grammar issues that were indicated.

This manuscript is a resubmission of an earlier submission. The following is a list of the peer review reports and author responses from that submission.

Round 1

Reviewer 1 Report

This is an outstanding article attempting to address differencees in social functioning and judgment for those with autism  vs. neurotypicals. Well written and comprehensive article. I do not have any changes to recommend.

Author Response

I am glad that reviewer found it relevant. 

Reviewer 2 Report

Evaluating a generalization task for homework during a training program of interpersonal problem-solving skills for adolescents and adults with autism spectrum disorder

General comments

  • Style and clarity: The language can be improved as there are numerous grammar, spelling and punctuation errors throughout the paper (e.g., “such us” line 39; “discreet” line 40). Additionally, some sentences (e.g., “By placing someone in a real situation…” lines 50-56) are hard to understand.
  • Design and methods: There is some room for clarification regarding the scoring and theoretical conceptualization of the worksheet.
  • Title and abstract: The title is a little too long. The abstract is concise, although it should also include details on the methodology (e.g., sample sizes and statistical analyses).
  • Analysis and sample size: The methodology of adopting non-parametric tests is appropriate for the small sample sizes.
  • Figures: A translated version of Figure 1 could be provided, although the authors’ remarks to append Figure 1 is also appropriate and translated items are included in the main text.

*The article’s focus on adolescents and adults may not be directly relevant to the scope of the journal.

Miscellaneous comments/questions:

  • ECI-Generalization task represented by the worksheet should be introduced in more detail in the Introduction section. As several theories are mentioned, an explanation of the relevant theory to ECI-Generalization can be included for more detail. The alignment of ESCI-Generalization and the objectives and skills taught in SCI-Labour should be explained as well to improve cohesion between theory, intervention, and measurement.
  • It appears that the dependent and independent variables for the logistic regression model are reversed. The dependent variable in a logistic regression model should be categorical, but the Total score identified as the dependent variable is continuous.
  • Discussion on validity of ESCI-Generalization is lacking as reliability is not the same as validity. A discussion of ESCI-Generalization in relation to the intervention (SCI-Labour) and any relevant theoretical background can be included instead.
  • It is not clear how IQ was assessed (section 2.1).
  • Why was ALT-CONT not included in the Total score?
  • Is there a reason why ALT-QLTY is heavily weighted in the total score over the other factors (total of 56 possible points out of 108)?
  • Please provide the precise inter-rater reliability statistic.
  • Please provide degrees of freedom in the Chi-square results (Table 3).
  • Please specify alpha level used in interpreting the statistical analyses.

Author Response

Q1-English language and style are fine/minor spell check required

Grammar, spelling and punctuation errors were corrected. A translator reviewed the full article

Q2-Clarify the scoring and theoretical conceptualization of the worksheet.

Theorical conceptualization is clarify in the Introduction section (Lines 68-72).

Q3-The title is a little too long. The abstract is concise, although it should also include details on the methodology (e.g., sample sizes and statistical analyses).

Title was shorten. Abstract was reviewed as required.

Q5-Figures: A translated version of Figure 1 could be provided, although the authors’ remarks to append Figure 1 is also appropriate and translated items are included in the main text.

Figure was translated (Post-treatment social situation was chosen as it is the common task to ASD Group and Control Group). In any case, both social situations are described in Table 1. It is still included as Figure not in Appendix. We are open to change it to the Appendix section in case editors consider it to present higher resolution.

Q6-ESCI-Generalization task should be introduced in more detail in the Introduction section. An explanation of the relevant theory to ESCI-Generalization can be included for more detail. The alignment of the task and the objectives taught in SCI-Labour should be explained.

A short explanation was included on the task’s theorical background (Lines 68-72). The alignment of the task and the objectives taught in the program are explicitly state in the section describing the program (2.2 Lines 93-103) and the target measure (2.3 lines 108-116 and 122 to126)

Q7-Revise the variables assigned for the logistic regression model.

There was a mistake in the statistic Design section that was corrected (lines 190-192)

Q8-A discussion of ESCI-Generalization in relation to the intervention (SCI-Labour) and any relevant theoretical background can be included.

Additional information was included in order to emphasize the correlation of task and intervention (lines 101-105 and 116-124)

Q9-It is not clear how IQ was assessed (section 2.1).

Description was completed (Line 75). It was fully described in the original study already published.

Q10-Why was ALT-CONT not included in the Total score?

Because ALT-CONT referred to Content of Alternatives. There was no punctuation associated with answers. Answers were classified based on the type of answer but in this study there was not range of different values among answers.

Q11-Is there a reason why ALT-QLTY is heavily weighted in the total score over the other factors (total of 56 possible points out of 108)?

The rationale behind the score is the same as for the rest of dimensions. The difference is that this dimension (Quality of Alternative) includes FOUR SUBDOMAINS (activity, relevancy, perspective and Quality of Action) to fully explore the quality of an alternative. When the answer was fully addressing each subdomain, 2 points, when partially addressed, 1 point. Finally, scores in four subdomains have to be summed for scoring

Q12-Please provide the precise inter-rater reliability statistic.

English writing was reviewed to describe the coding reliability process (Lines 174-180). We hope it is clearer now:

Inter-rater reliability was assumed once two raters (blind study hypothesis) were trained and obtained more than 80% of agreement in 28 worksheets. After they agree, discussing the difference between codes in each dimension and subdomains; then one single rater coded the full sample of homework task.

Q13-Please provide degrees of freedom in the Chi-square results (Table 3).

Degrees of freedom is now included in tables

Q14-Please specify alpha level used in interpreting the statistical analyses.

Table 3, 4 and 6 indicate the exact alpha level (significant difference when p<.001). Table 5 and 7 include notes to specify alpha level (p<.001).

Reviewer 3 Report

Comments to the author:

The current study tested the effectiveness of the ESCI-Generalization Task as a performance measure of social problem-solving skills in a sample of adolescents and adults with ASD. I believe that the current work provides some very promising evidence for the intervention for adolescents and adults with ASD. One major point of confusions is whether the CG received the intervention. If it did, pre- and post- scores should have been incorporated into the analyses, so I am guessing that they did not receive the intervention. In this case, also, that is a major limitation that should be addressed. It could be that the ASD group was simply better practiced at completing the tasks and that is why the decrease in effect size was seen.

Method

  • In the participants section, it states: “The sample was composed of 44 participants (42 men and 6 women) with ASD…The comparison group (CG) were 48 subjects (43 men and 7 women).” Those numbers don’t add up: 42 + 6 = 48 and 43+7 = 50.
  • In the participants section, it states that the mean IQ of the ASD group was 96.27 and the mean IQ of the comparison group was 103.75. It also says that the groups were matched on verbal IQ standard scores. Could the authors talk a little more about how the groups were matched? Was the CG recruited to match the ASD group on IQ? Was an independent samples t-test done to confirm that there was no group difference in IQ (96.27 compared to 103.75)? If not, I would recommend this be done.
  • The numbering of the sessions on page 3-4, lines 91-96 are either off or difficult to follow.
  • In section 2.2 – It says that “The original SCI-Labour Programme is a 10-week (90 min sessions) interpersonal 85 problem-solving training (Calero, García-Martín & Bonete, 2019) adapt for adolescents 86 and adults with ASD” – thus, is it also appropriate to offer to a control group? For example, because it is adapted for ASD, the results may be skewed to reflect more growth for the participants with ASD, conflating findings. The final sentence in this section states: “A previous study supported the feasibility and effectiveness through an open trial (Bonete, Calero & Fernández-Parra, 2015)” but did this study support feasibility and effectiveness in non-ASD groups? Or, perhaps I am confused as later on, on page 7, it states that only post scores were used for the cg. Does that mean that the cg never received the intervention? I think this could be more clear up front.

Results

  • The first few lines on page 8 (lines 199-201): “When comparing Pre and Post in ASD-Group, it was clearly observed that ASD- 199 Group improved in giving more complete responses concerning the Problem Definition, 200 the Quality of the possible causes and the Solution Suitability” would fit more appropriately in the discussion. The results in the table only show the comparison between the ASD post-intervention scores and the CG – it doesn’t seem to directly compare pre- and post-scores within the ASD group. Any comment on comparisons between Table 3 and 4 should be in the discussion, not results.
  • Although not the time points of focus in the current study, it may be helpful to report mean scores at each time point (3-10) if they are coded, just to contextualize the changes in the ASD group over time.

Discussion

  • There is a typo on line 283: “Pos-treatment” should be “Post-treatment”
  • In addition to measuring if changes were sustained (e.g., follow up), future directions for this work could include looking at each time point and see when changes are occurring. For example, are the achieved benefits seen after just 6 sessions? If so, the intervention may be able to be modified and made more broadly available, maybe with the addition of a booster session or two if effects are not maintained.
  • I’m still confused as to whether the CG received the intervention. On page 14, lines 333-334: “Another interesting line of research would be to validate the ESCI-Generalization Task in a wide neurotipical sample, who received the program as well” – are the authors saying that the cg in this future study could give the program to a neurotypical sample or are they saying that the current one did? If it did, pre- and post- scores should have been incorporated into the analyses, so I am guessing that they did not receive the intervention. In this case, also, that is a major limitation. It could be that the ASD group was simply better practiced at completing the tasks. If both groups had gotten the intervention, the effect sizes between the groups may have been the same pre- and post-intervention.
  • One point that is missing from the discussion is how these skills will generalize to real-world skills in the workplace and in daily life, which, I think, is the whole point of the intervention. I would encourage authors to bring this into the paper throughout the intro and discussion.

Author Response

Q1-Clarify if the CG received the intervention. It is a major limitation that should be addressed as it could be that the ASD group was simply better practiced at completing the tasks and that is why the decrease in effect size was seen

CG did not receive the intervention. It is now clearly specified.

We agree that it is a limitation that will be reflected in the limitations section.

Q2-In the participants section, it states: “The sample was composed of 44 participants (42 men and 6 women) with ASD…The comparison group (CG) were 48 subjects (43 men and 7 women).” Those numbers don’t add up: 42 + 6 = 48 and 43+7 = 50.

We apologize there were a mistake, including data from full sample (original study). This is now corrected for the participants included in this analysis (only participants who delivered Pre ESCI-Generalization task and/or Post ESCI-Generalization task.

Q3-In the participants section, it states that the mean IQ of the ASD group was 96.27 and the mean IQ of the comparison group was 103.75. It also says that the groups were matched on verbal IQ standard scores. Could the authors talk a little more about how the groups were matched? Was the CG recruited to match the ASD group on IQ? Was an independent samples t-test done to confirm that there was no group difference in IQ (96.27 compared to 103.75)? If not, I would recommend this be done.

Exactly, the CG was recruited to match the sample of ASD group on sex, age and IQ in the original study in which 50 participants were enrolled. However, only 44 participants fulfill one of the two tasks (Pre or Post) or both. That is why numbers change. Specifically, only 37 completed PRE ESCI-Generalization task. Five participants did not fulfill the PRE ESCI-Generalization task; and 39 completed POST ESCI-Generalization task, they were other different five participants that did not fulfill the POST ESCI-Generalization task. That is the reason why we decided to include the full sample of control group that answered the POST-ESCI-Generalization task.

IQ scores were compared for this final sample, and they were equivalent in verbal IQ, sex and age so we considered comparable groups as the original study.

Q4-The numbering of the sessions on page 3-4, lines 91-96 are either off or difficult to follow.

Changes were done for clarifying

Q5-The open trial Bonete, Calero & Fernández-Parra, 2014 support feasibility and effectiveness in non-ASD groups? If it is adapted to ASD, the results may be skewed to reflect more growth for the participants with ASD, conflatin findings.

Clear up if CG received intervention.

CG did not receive the intervention. Feasibility and effectiveness of this program is tested in different populations through different studies (Ed. García-Martín & Calero, 2019[12]) but in this study, it was not implemented to the control group sample.

Q6-Lines 199-201 on page 8 would fit more appropriately in the discussion

We believe these statements must be included in the Result section as we are still summarizing when tables show. In the Discussion section we also highlight this results in general terms 

Q7-The results in the table only show the comparison between the ASD post-intervention scores and the CG – it doesn’t seem to directly compare pre- and post-scores within the ASD group. Any comment on comparisons between Table 3 and 4 should be in the discussion, not results.

Pre- and Post- scores within the ASD group are included in tables 6 and 7. Comments on comparisons were reviewed and vocabulary was adjusted to avoid data interpretation.

Q8-Although not the time points of focus in the current study, it may be helpful to report mean scores at each time point (3-10) if they are coded, just to contextualize the changes in the ASD group over time.

The improvements of the ASD group due to the program are addressed in other paper. We now include this information and reference in line 300.

Q9-There is a typo on line 283: “Pos-treatment” should be “Post-treatment”

Corrected

Q10-I’m still confused as to whether the CG received the intervention. On page 14, lines 333-334: “Another interesting line of research would be to validate the ESCI-Generalization Task in a wide neurotypical sample, who received the program as well” – are the authors saying that the cg in this future study could give the program to a neurotypical sample or are they saying that the current one did? If it did, pre- and post- scores should have been incorporated into the analyses, so I am guessing that they did not receive the intervention. In this case, also, that is a major limitation. It could be that the ASD group was simply better practiced at completing the tasks. If both groups had gotten the intervention, the effect sizes between the groups may have been the same pre- and post-intervention.

It is clearer now that CG did not receive intervention so that is why it is suggested as a future line of research in order to wider the sample of people with normal development and explore program’s impact, specially concerning the cross-cultural validity. We mention this review suggestion in line 340

Q11-One point that is missing from the discussion is how these skills will generalize to real-world skills in the workplace and in daily life, which, I think, is the whole point of the intervention. I would encourage authors to bring this into the paper throughout the intro and discussion.

We included some reference along the Discusion section  with relevant references (Line 339)